# Wnt ligands influence tumour initiation by controlling the number of intestinal stem cells

D.J. Huels[1], L. Bruens[2,3,4], M.C. Hodder[1], P. Cammareri[1], A.D. Campbell[1], R.A. Ridgway[1], D.M. Gay[1], M. Solar-Abboud[1], W.J. Faller [1], C. Nixon[1], L.B. Zeiger[1], M.E. McLaughlin[5], E. Morrissey[6], D.J. Winton[7], H.J. Snippert[4], J. van Rheenen[2,3] & O.J. Sansom [1,8]

Many epithelial stem cell populations follow a pattern of stochastic stem cell divisions called 'neutral drift'. It is hypothesised that neutral competition between stem cells protects against the acquisition of deleterious mutations. Here we use a Porcupine inhibitor to reduce Wnt secretion at a dose where intestinal homoeostasis is maintained despite a reduction of Lgr5+ stem cells. Functionally, there is a marked acceleration in monoclonal conversion, so that crypts become rapidly derived from a single stem cell. Stem cells located further from the base are lost and the pool of competing stem cells is reduced. We tested whether this loss of stem cell competition would modify tumorigenesis. Reduction of Wnt ligand secretion accelerates fixation of *Apc*-deficient cells within the crypt leading to accelerated tumorigenesis. Therefore, ligand-based Wnt signalling influences the number of stem cells, fixation speed of *Apc* mutations and the speed and likelihood of adenoma formation.

[1] CRUK Beatson Institute, Glasgow G61 1BD, UK. [2] Hubrecht Institute, Royal Netherlands Academy of Arts and Sciences (KNAW) and UMC Utrecht, 3584 CT Utrecht, The Netherlands. [3] Molecular Pathology, Oncode Institute, The Netherlands Cancer Institute, 1066CX Amsterdam, The Netherlands. [4] Center for Molecular Medicine, Oncode Institute, University Medical Center Utrecht, 3584 CG Utrecht, The Netherlands. [5] Oncology Translational Research, Novartis Institutes for Biomedical Research, Cambridge, MA 02139, USA. [6] MRC Weatherall Institute of Molecular Medicine University of Oxford, John Radcliffe Hospital, Headington, Oxford OX3 9DS, UK. [7] CRUK Cambridge Institute, Cambridge CB2 0RE, UK. [8] Institute of Cancer Sciences (ICS), University of Glasgow, Glasgow G12 8QQ, UK. Correspondence and requests for materials should be addressed to O.J.S. (email: o.sansom@beatson.gla.ac.uk)

The intestinal epithelium is constantly renewing and new epithelial cells are continuously produced by a small number of intestinal stem cells (ISCs) located at the base of the crypt[1, 2]. These crypt columnar stem cells exhibit the highest levels of Wnt signalling demonstrated by nuclear β-catenin staining and high expression of a number of Wnt target genes including Lgr5. The expression of Lgr5 within these cells amplifies Wnt ligand signalling as R-spondin binds to the LGR5 receptor to agonise Wnt signalling[3]. Wnt ligands are produced by the epithelium (Wnt3 from the Paneth cells) and the mesenchyme[4], which also produces R-spondin. Although excellent evidence exists showing that Lgr5+ cells can act as functional stem cells in the adult intestinal epithelium, these cells are dispensable for homoeostasis (over the measured time period of 10 days), though they are required for regeneration post irradiation[5, 6].

Over recent years it has been accepted that these Lgr5 ISCs are not long lived but rather replace each other in a stochastic fashion termed 'neutral drift'. Functionally, the progeny of one stem cell displaces all other stem cells from the niche and becomes fixed in the crypt[7, 8]. This stochastic replacement also explains stem cell dynamics in other proliferating tissues like skin or during spermatogenesis[9].

Through this process it is hypothesised that cells that have acquired deleterious mutations would be likely displaced by their neighbouring wild-type ISCs. In vivo imaging studies have monitored this competition in real time. These studies have shown that stem cells at border regions (higher up from the base of the crypt) can be pushed into the transit amplifying zone of the crypt, while stem cells at the centre/base of the crypt are more likely to be retained. Importantly border stem cells can also return to the centre position and regain functional stem cell properties[1].

The Wnt signalling pathway has been shown to be required for intestinal homoeostasis. Genetic loss of the Wnt transcription factor Tcf4/β-catenin or suppression of Wnt ligand-based signalling via Dkk overexpression led to rapid loss of small intestinal crypts[10–12]. Additionally, R-spondins are essential for ISCs and crypt maintenance, since complete inhibition by blocking their Lgr5 and Znrf3/Rnf43 binding domains results in crypt death. However, inhibition of only one of the binding domains results in a reduced number of Lgr5+ cells but otherwise normal crypt homoeostasis. This reduction in Lgr5+ cells resulted in a rapid fixation of the remaining ISCs in the crypt[13].

This model where stem cells compete with each other has also been applied to examine oncogenic events that might confer advantages to ISCs carrying that mutation. Common mutations in CRC (Kras^{G12D} mutation and Apc deletion) have been targeted to ISCs in the mouse and have been shown to influence the neutral drift. In both cases there was a greater chance for these mutated stem cells to replace other wild-type cells in the crypt[14, 15].

One technical caveat to these studies is the methodology used to mark crypts in the mouse and determine ISC dynamics. The studies comparing the advantages of a specific mutation relied on cre-mediated expression of KRAS^{G12D} mutation or Apc gene deletion within ISCs. Tracking of mutant clones were all performed using a reporter gene from the Rosa26 promoter (e.g. Rosa26-Lox-Stop-Lox-tdTomato) allele rather than assaying recombination at the Kras or Apc locus itself. Importantly previous studies have suggested there may be discordance between reporter alleles[16]; therefore, there is a possibility that the precise rates of advantages of these alleles may be different if there are clones that fail to recombine the gene of interest and only the reporter gene (and vice versa). Moreover, it is important to realise that even if mutations are non-neutral, these mutations are not deterministic: mutated stem cells are still surrounded by non-mutated stem cells and therefore have a high chance to be replaced by wild-type cells.

Instead of complete inhibition, reduction of Wnt signalling by Wnt ligand inhibitors in adult mice and humans has shown that these inhibitors can be well tolerated, though do result in dramatically fewer stem cells[17]. Here we have used a Porcupine inhibitor to reduce Wnt secretion and test the consequences of lowering the number of functional stem cells per crypt and its effect on stem cell competition. We find that crypts are rapidly derived from a single stem cell and using in vivo imaging we show the reason for this is that stem cells at border regions are rapidly lost. To test the functional relevance of this reduction of stem cell competition for tumorigenesis, we examined whether this would accelerate the fixation of Apc-deficient cells in the intestine. Importantly, we discovered that the widely used R26tdTomato reporter poorly reports efficient recombination of the two Apc alleles in vivo. Using RNA in situ and immunohistochemistry to report loss of Apc we show that reduction of Wnt ligands results in more efficient tumorigenesis due to the rapid fixation of Apc-deficient crypts.

Together our data suggest neutral drift and stem cell competition require an optimal level of ligand driven Wnt signalling that has evolved to allow the rapid loss of deleterious mutation but also to protect against the acquisition of advantageous tumour-promoting mutations.

## Results

**Wnt inhibition only has minor effects on homoeostasis.** We assessed intestinal homoeostasis after treatment with a Porcupine inhibitor LGK974, otherwise known as WNT974[17]. Consistent with previous studies targeting Porcupine and other strategies to dampen Wnt ligand-based signalling[18], reduction of Wnt ligand secretion did not change rates of crypt proliferation (Fig. 1a, b). Although the total number of proliferating cells per crypt did not change, there was a small yet significant alteration in the distribution of the proliferative cells. Following porcupine treatment all the proliferative cells are found lower in the crypt compared to the vehicle/untreated mice (Supplementary Figure 1). The most striking impact of Wnt inhibition was the marked down-regulation of several ISC genes, e.g. Lgr5, Olfm4 and Lrig1 (Fig. 1a, c). Interestingly, there was no change in Bmi1 expression (Fig. 1c). We performed RNA sequencing from whole intestine to analyse the effect of reduced Wnt signalling on global gene expression (Supplementary Figure 2a and Supplementary Data 1), which revealed only a small number of significantly deregulated genes (22 upregulated, 44 downregulated). Among the downregulated genes was the canonical Wnt target gene Axin2, and the ISC genes Olfm4 and CD133. We observed a reduced number of Paneth cells only after long-term treatment (~30 days), which indicates the requirement of Wnt ligands for the generation of new Paneth cells (Supplementary Figure 2b).

Given this relatively mild effect on homoeostasis we wanted to test if intestinal regeneration following irradiation was affected as this is a Wnt regulated process and requires Lgr5+ cells[11, 19]. We observed an inhibition of intestinal regeneration after irradiation (Supplementary Figure 2c) consistent with previous studies[18]. We also investigated whether Porcupine inhibition could suppress hyperproliferation and increase crypt number induced by oncogenic mutations (either Braf^{V600E/+} or Braf^{V600E/+} Pten^{fl/fl}) and again saw a marked impact on these phenotypes (Supplementary Figure 3). Finally given the previous genetic studies showing that complete Wnt ligand inhibition caused the loss of crypts[12], we examined if Porcupine inhibition in combination with reduced β-catenin levels might disrupt tissue homoeostasis. The Porcupine inhibitor at the used dosage is well tolerated and

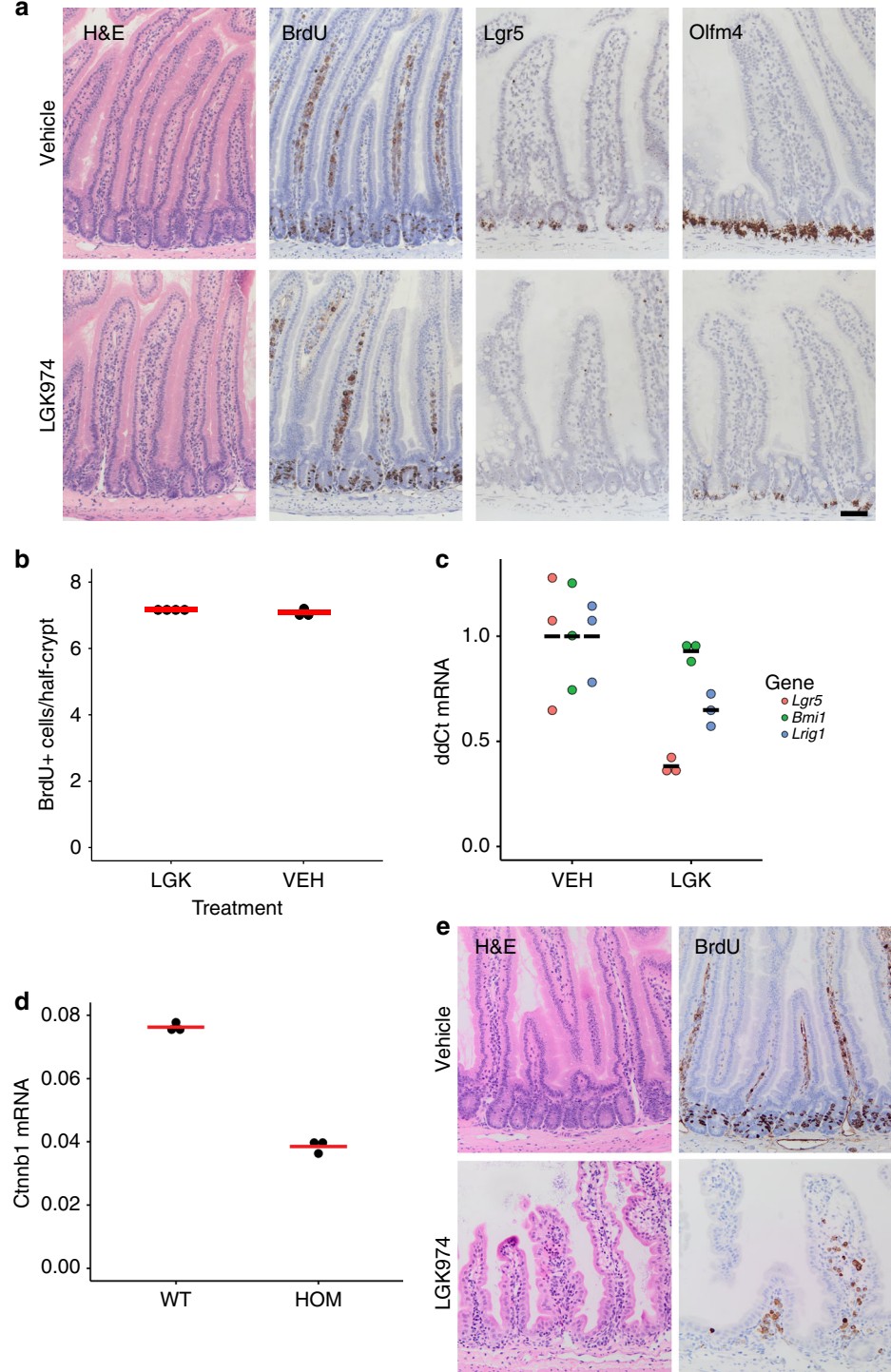

**Fig. 1** Homoeostasis is unperturbed after LGK974 treatment. **a** C57BL/6 mice were treated with LGK974 for 4.5 days. The small intestine showed no toxicity and no difference in proliferation (BrdU). Treatment of LGK974 leads to downregulation of the intestinal stem cell genes *Lgr5* and *Olfm4* as confirmed by RNA in situ hybridisation. Note only a few cells at the bottom of the crypt still express *Olfm4* after LGK974 treatment (*N* = 3 for both groups). **b** Quantification of BrdU+ cells/half-crypt (at least 30 crypts per mouse were analysed). Each dot represents the average per mouse, red bar = mean per group. Vehicle (VEH) *N* = 3, LGK974 (LGK) *N* = 4. **c** qRT-PCR confirms downregulation of several stem cell genes, whereas expression of *Bmi1* is unchanged. Each dot represents single mouse sample, black bar indicates mean per group, *N* = 3 per group. **d** Uninduced *Catnb*^lox(ex3)/lox(ex3) are hypomorphs with about 50% reduced expression of β-catenin (*Ctnnb1*), as confirmed by qRT-PCR (*N* = 3). **e** Reduced expression of β-catenin results in hyper-sensitivity to LGK974 and loss of the (small) intestinal crypts within 10 days (mean survival), *N* = 8. Scale bar = 50 μm

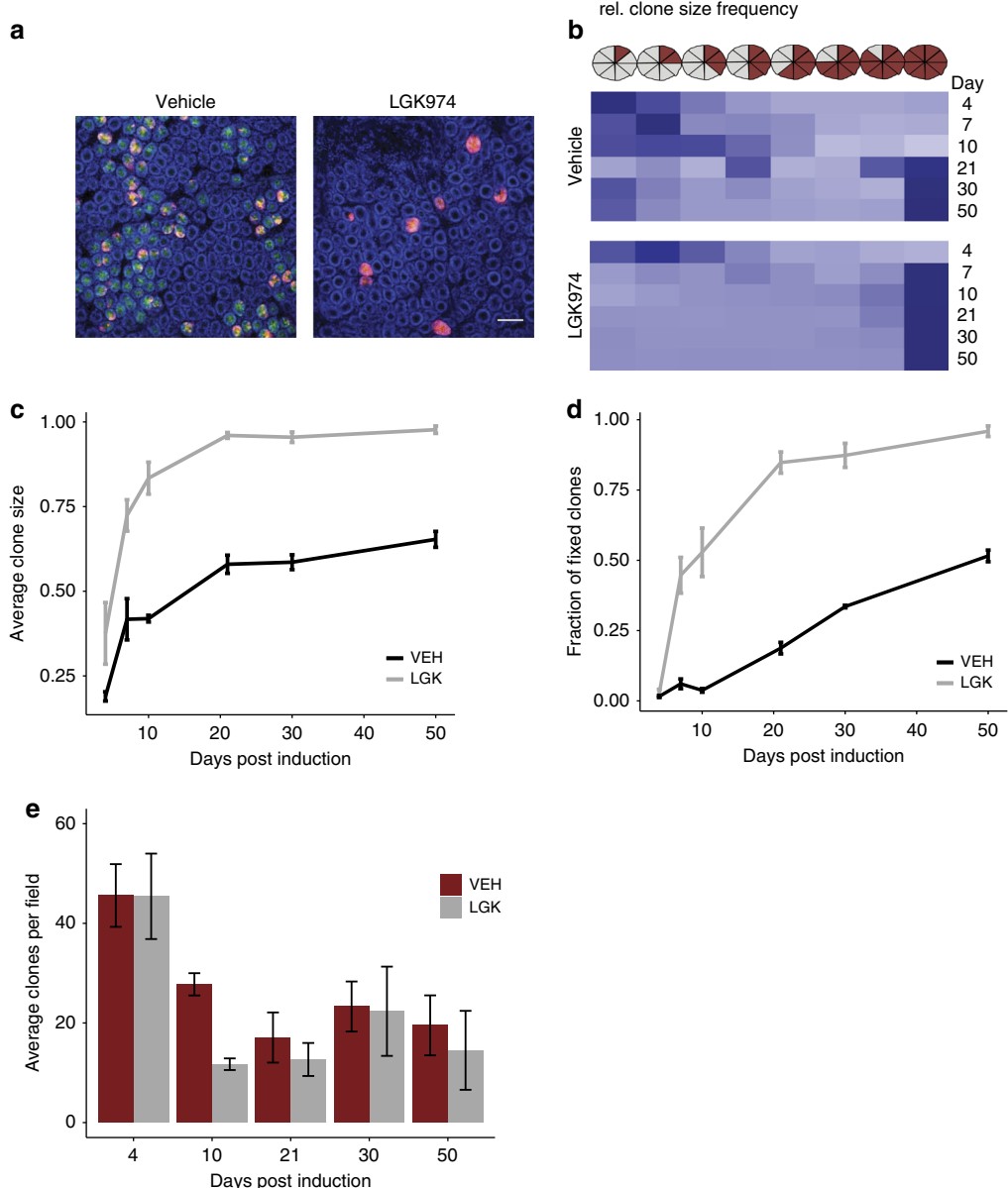

**Fig. 2** Stem cell replacement rate is accelerated after LGK974 treatment. Mice were induced with 0.15 mg tamoxifen to induce tdTom$^{fl}$ recombination in few *Lgr5-CreER-EGFP*+ cells. **a** Mosaic expression of the Lgr5-eGFP (green) and the recombined tdTom$^{fl}$+ cells (red). Nuclei were stained with DAPI (blue). Representative pictures at day 10, note the loss of GFP expression and clones are fully labelled by tdTom+ cells in Porcupine inhibitor-treated mice (LGK974). Scale bar = 100 um. **b** Clone size was counted in 'eighths', at time indicated after induction. At least 200 clones per mouse were counted, vehicle (VEH) $N$ = 3, 3, 4, 4, 3, 3 and LGK974 (LGK) $N$ = 2, 4, 3, 5, 3, 3 for each timepoint, respectively. Heatmap shows all counted clones per timepoint/ group. Note that LGK974 has an increased clone size at day 4 and the mean clone size from day 7 is almost at its maximum. Graph shows the mean clone size (**c**) or the number of fully fixed crypts (**d**) at different time points as shown in **b**. Error bars, s.e.m. **e** Number of tdTom+ clones per field. The number of crypts with at least one tdTom+ cell were counted per field, ≥19 images per mouse, error bars = s.e.m. Note the similar number of clones at day 4 but the greater reduction in clones after LGK974 treatment at day 10. Vehicle (VEH) $N$ = 3, 3, 4, 4, 3, 3 and LGK974 (LGK) $N$ = 2, 4, 3, 5, 3, 3 for each timepoint respectively

mice can be treated for several weeks. Mice with reduced expression of β-catenin are viable and display no phenotype throughout their life. However, a reduction of β-catenin expression by 50% renders these mice sensitive to the Porcupine inhibitor and resulted in rapid crypt loss within 8–12 days (Fig. 1d, e). Together these data highlight that the intestinal epithelium can tolerate a reduction in Wnt ligand signalling but further reduction of β-catenin causes a complete loss of crypts. This is consistent with previous work using higher doses of Porcupine inhibitors[17, 18].

**Reduced Wnt ligand secretion decreases crypt fixation time.** Given the very selective effect of Wnt ligand reduction via Porcupine inhibition upon *Lgr5* and other ISC genes, we wanted to assess the impact this had upon ISC dynamics. To do this we used well-established techniques examining the ability of single stem cells (labelled by tomato) to repopulate entire crypts.

This was achieved using the *Lgr5-EGFP-Cre$^{ER}$* (*Lgr5Cre$^{ER}$*) mouse crossed to the *R26R-LoxStopLox-tdTomato* (*tdTom$^{fl}$*) mouse. Induction with a previously established low tamoxifen concentration resulted in recombination of very few *Lgr5*+ cells

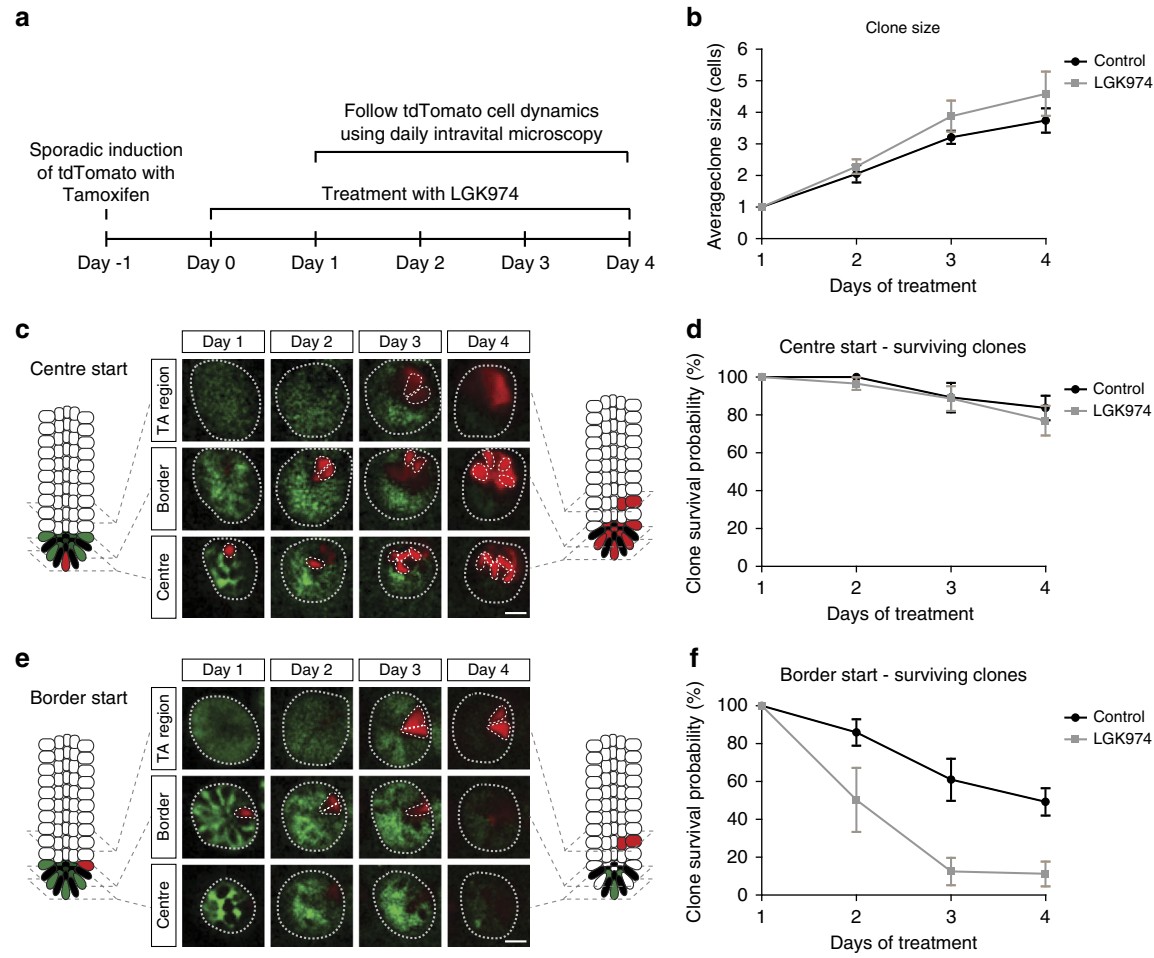

**Fig. 3** In vivo live imaging shows specific loss of border stem cells after LGK974 treatment. **a** Graphical representation of the experimental setup. One day before LGK974 treatment, *Lgr5Cre*[ER] *tdTom*[fl] mice were injected with 0.05 mg tamoxifen IP to induce recombination in single cells. Mice (*N* = 4) were treated with LGK974 and daily intravital imaging was performed starting 1 day after first LGK974 treatment and compared to control mice (*N* = 5). **b** Graph shows mean clone size (43 crypts, control; 56 crypts, LGK974 on day 1) of surviving clones (clones with at least one cell in centre or border) over time within the centre and border. Note that cells in the transit amplifying (TA) cell region are not counted. **c**, **e** Intravital images of the same crypt on days 1–4 following a clone starting in the centre (**c**) and a clone starting in the border (**e**). **d**, **f** Graphs show the percentage of clones that still have at least one cell in the centre or border starting from centre cell (17 crypts, control; 36 crypts, LGK974) (**d**) or from border cell (26 crypts, control; 20 crypts, LGK974) (**f**). Error bars = s.e.m. Scale bar, 20 μm

per crypt which are then permanently labelled by expression of the tdTom reporter. The fate of these stem cells can be monitored to see if they are lost or take over the full crypt. *Lgr5Cre*[ER] *tdTom*[fl] mice were induced and treated from 24 h post induction with either LGK974 or vehicle. After treatment with LGK974 we observed a reduction in the Lgr5-GFP signal consistent with our previous qRT-PCR and ISH data (Fig. 2a). Importantly, we saw a dramatic increase in the average clone size after administration of the Porcupine inhibitor. This was observable 4 days post induction and further increased during the time course (Fig. 2b, c). This resulted in a striking increase in the number of fully fixed clones (Fig. 2d), with more than 80% of crypts fully fixed after 3 weeks (<20% in the vehicle control), a process which usually takes about 2–4 months[7, 8]. The competition between labelled and unlabelled cells within a crypt resulted in many crypts losing the tdTom label, in accordance with neutral drift. The vehicle treatment resulted in a progressive decline in the number of tdTom+ crypts, whereas treatment with LGK974 accelerated this process and the final number of tdTom+ crypts

was reached after 10 days (Fig. 2e). This analysis also confirmed that an equal number of crypts were recombined before the start of the treatment (Fig. 2e, day 4). To investigate clonal dynamics using a system that is not limited to recombination in only Lgr5-positive stem cells, we repeated this experiment using another inducible cre (*AhCre*[ER], not driven by *Lgr5*), which has previously been used for stem cell dynamic studies[14]. Using an established low dose induction of *AhCre*[ER] mice, crossed to the *Rosa26 tdTom*[fl] mice, we observed a similar increase in the average clone size after LGK974 treatment compared to vehicle treatment 10 days after induction (Supplementary Figure 4a).

**Border stem cells are lost after reduction of Wnt ligand secretion**. The dramatic increase in clone size at early time points following Porcupine inhibitor treatment gave us an excellent opportunity to image the *Lgr5Cre*[ER] *tdTom*[fl] mice in vivo to elucidate why the stem cell dynamics were altered. The in situ hybridisation analysis for *Lgr5* and *Olfm4* showed that there is a reduction in the expression of these ISC markers. These data

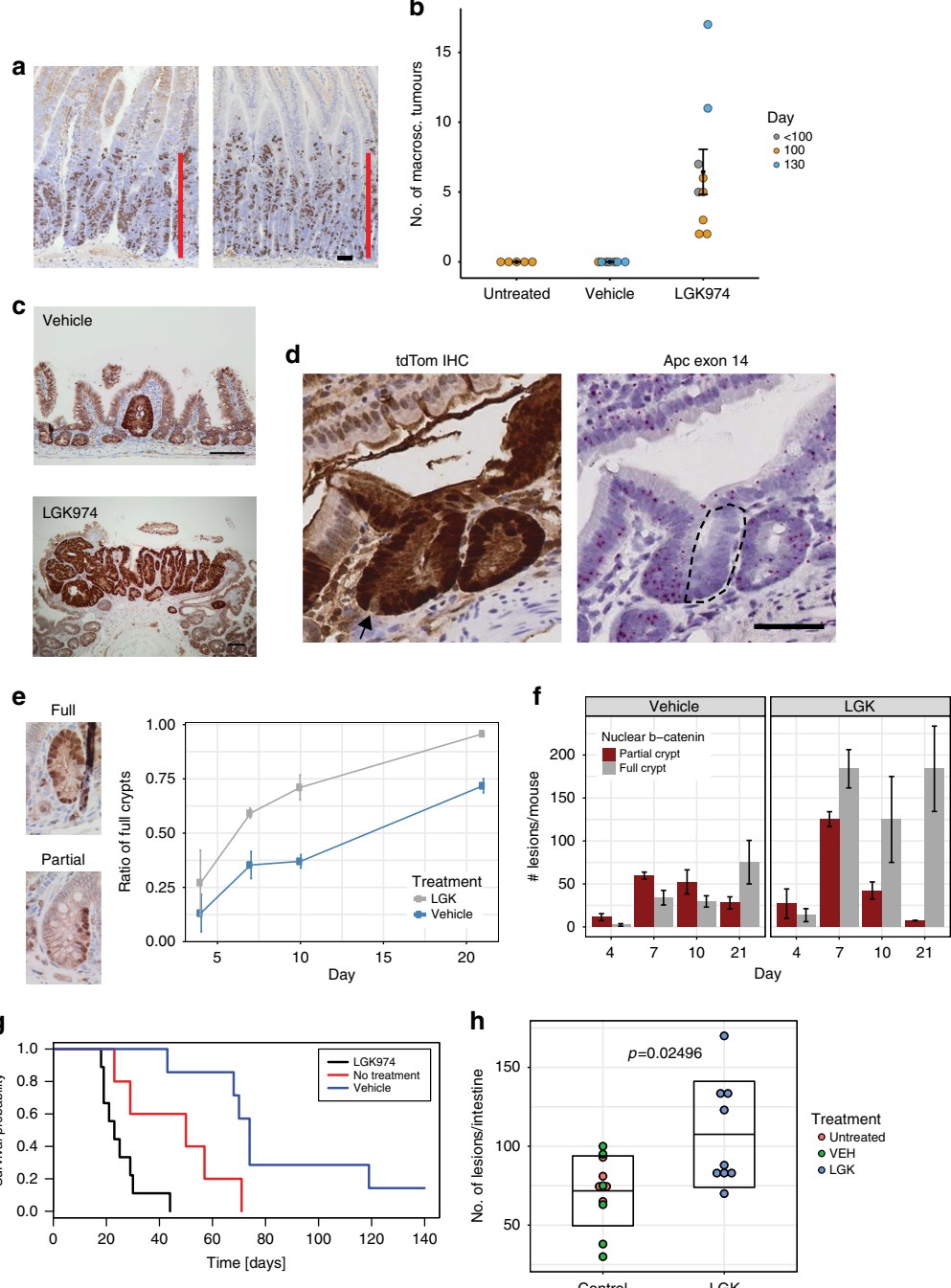

**Fig. 4** Reduced Wnt ligand secretion does not affect *Apc*-deficient cell growth but accelerates tumorigenesis. **a** *VilCre^ER^ Apc^fl/fl^* mice were induced and treated with LGK974 starting the following day. Mice were sampled at day 4 post induction, and proliferation assessed by BrdU staining. Scale bar = 50 µm. **b** *Lgr5Cre^ER^ Apc^fl/fl^* mice induced with 0.15 mg tamoxifen and treated with LGK974/vehicle 1 day p.i. The number of macroscopic adenomas was scored when mice showed signs of intestinal adenomas or at 100/130 day timepoint. Each dot represents one mouse, untreated N = 5, vehicle N = 7 and LGK974 N = 9, black dot = mean, error bars = s.e.m., Mann–Whitney U test: untreated vs. LGK974 p = 0.002639, vehicle vs. LGK974 p = 0.0006014. **c** Immunohistochemistry for β-catenin showed only small microadenomas in the vehicle group, in contrast to adenomas found after LGK974 treatment. Scale bar = 100 µm. **d** Example image of *Lgr5Cre^ER^ Apc^fl/fl^ tdTom^fl/+^* mouse 10 days post induction (3 mg, LGK974 treatment). Immunohistochemistry for tdTomato (RFP) shows fully labelled crypt. Arrow marks unlabelled cell (probably Paneth cell), suggesting a newly labelled crypt. *RNA* in situ for Apc exon 14 shows that only half of the crypt has recombined (dashed area). Scale bar = 50 µm. **e** *Lgr5CreER Apc^fl/fl^* mice were induced with 3 mg tamoxifen and treated with LGK974/vehicle starting at day 1 p.i. Crypts were scored based on immunohistochemistry for β-catenin and categorised into partial and full crypts. The ratio of full crypts is in relation to the sum of full and partial crypts. **f** Scoring of partial and full crypts at different timpoints in absolute numbers. N = 3 mice for each timepoint and each group, error bars = s.e.m. **g** *Lgr5CreER Apc^fl/fl^* mice induced with 3 mg tamoxifen were treated with LGK974 or vehicle starting at day 1 p.i. Mice were sampled when signs of intestinal tumour burden were apparent. Untreated N = 5, vehicle (VEH) N = 7, LGK974 N = 9 mice, log-rank test: vehicle vs. LGK974 p = 0.00019, untreated vs. LGK974 p = 0.0222. **h** Number of adenomas scored on histological sections. Untreated N = 5, vehicle (VEH) N = 6, LGK974 N = 9 mice. Each dot represents number of lesions per mouse; box indicates mean ± standard deviation. Mann–Whitney U test for LGK974 vs. control mice (untreated and vehicle) p = 0.02496

imply that either ISCs reduce expression of these genes, while still participating in the competition, or that there is a decrease in the number of functional stem cells per crypt.

To investigate this, we tracked the fate of single Lgr5+ cells via multiphoton intravital microscopy through an abdominal imaging window[1]. The $Lgr5Cre^{ER} tdTom^{fl}$ mice were induced with a low dose of tamoxifen and tdTom+ clones were followed over a period of 4 days (Fig. 3a). On average, an increase of clonal progeny (tdTom+) derived from the Lgr5+ cells was observed in both treatment groups with similar kinetics (Fig. 3b). It has been previously established that the fate of individual stem cells is determined by its position in the crypt. Depending on the position, Lgr5+ cells can be grouped as 'border stem cells' at the upper part of the stem cell niche and 'centre stem cells' positioned at the bottom of the crypt[1]. Despite loss of Lgr5-GFP expression after LGK974 treatment, originally labelled centre stem cells persisted at the centre and proliferated similar to the control cells (Fig. 3c, d). In contrast, following the fate of labelled clones at the border of the stem cell niche, we observed a reduction in the number of tdTom+ clones that remained after treatment with LGK974 (Fig. 3e, f). This suggests that reduction of Wnt ligand secretion leads to specific loss of stem cell activity at the upper part of the stem cell niche. The cells at the bottom of the crypt show no difference in clonal growth, despite downregulation of the ISC gene Lgr5. Thus, Wnt ligand inhibition is reducing the number of stem cells in the niche. Due to the resulting decreased competition, cells at the centre have a higher chance of repopulating the crypt quickly when compared with vehicle or untreated mice.

**Reduced stem cell pool facilitates adenoma formation**. The neutral drift of the ISCs is defined by two parameters, the number of stem cells and the stem cell replacement rate[14]. We observed that reduction of Wnt ligand secretion led to a reduction in the number of stem cells, namely the border stem cells, but otherwise normal kinetics of the centre stem cells. This is accompanied by an acceleration of single clones to become fixed. We next examined if these changes have consequences for tumour initiation.

Colorectal cancer is characterised by loss of the tumour suppressor gene APC and our previous studies have shown that this is sufficient to result in Wnt deregulation and adenoma initiation in vivo[20]. Importantly, organoid cultures from Apc-deficient cells grow as spheres and do not require R-Spondin, suggesting they are independent of Wnt ligand[21]. We tested if Apc-deficient cells are truly independent of secreted Wnt ligands in vivo. $VillinCre^{ER} APC^{fl/fl}$ mice were induced and treated with either vehicle or LGK974. As expected, we saw no impact of the crypt progenitor phenotype after LGK974 treatment, with marked hyperproliferation in both vehicle and LGK974-treated mice (Fig. 4a, Supplementary Figure 4b).

It has been reported that certain mutations (e.g. $Kras^{G12D/+}$ or $Apc^{-/+}$) impart an advantage on ISCs when compared with their neighbouring wild-type stem cells. This advantage is reflected in an increased probability for a mutant stem cell to replace its neighbour and ultimately become fixed so that all cells in the crypt derive from the same mutant clone[14, 15]. Our data suggest that a reduction of the stem cell pool also led to a dramatic acceleration in the time it takes for crypts to become monoclonal. This leads to the prediction that LGK974 might accelerate the rate of adenoma initiation through altering stem cell dynamics, despite not having an impact on Apc-deficient cells.

Previous studies have assumed an equivalent level of recombination between the R26R-LSL-tdTomato and the gene of interest. However in our experience we only obtain a robust

tumorigenic phenotype in $Lgr5Cre^{ER} Apc^{fl/fl}$ mice with a single injection of high concentration tamoxifen[22] ($\geq 2$ mg tamoxifen, Supplementary Figure 4c). We therefore first decided to test whether LGK974 could lead to tumour formation following low-level deletion (0.15 mg) in $Lgr5Cre^{ER} Apc^{fl/fl}$. Mice were induced with the low-level tamoxifen and treated 24 h after induction, which assures recombination of the same number of cells before treatment. We observed that LGK974 treatment resulted in a number of macroscopic adenomas, whereas none of the vehicle or untreated mice had any visible adenomas (Fig. 4b). We then examined if microscopically there were more lesions on the intestines dissected from these mice. These can be visualised by immunohistochemistry for β-catenin (as Apc is deleted). Histological analysis of an intestinal 'swiss roll' showed that the majority of vehicle and untreated mice had no lesions (3/5 and 4/7, respectively) with the other mice containing only a single microadenoma. In contrast, almost all the LGK974-treated mice had several adenomas per section (6/7 mice, Fig. 4c). These data suggest that Porcupine inhibition does affect tumorigenesis in this murine model of colorectal cancer.

However, the low number of histological lesions in the control mice did not match the high number of recombined tdTom+ crypts, we would expect based on our clonal analysis with the same low tamoxifen induction (Fig. 2e). We thought of two possibilities that could explain this discrepancy: 1. both copies of Apc are deleted in a large number of crypts, but only few of them accumulate β-catenin and progress to adenoma formation; 2. that loss of Apc was occurring at a much lower frequency than the tdTom reporter would suggest.

To test the latter possibility, we utilised the latest in situ hybridisation technology 'Basescope'. This technology allows accurate detection of RNA transcripts by use of short RNA detection probes. We designed probes that enabled the detection of exon 14 (117 bp) of Apc, which is deleted following recombination. We confirmed the specificity of the RNA in situ probes on established adenomas from $Lgr5Cre^{ER} Apc^{fl/fl}$ mice, both for the unrecombined and the recombined Apc allele (Supplementary Figure 5a, b). Using these probes, we examined the level of Apc deletion after a single injection of high concentration tamoxifen (3 mg $Lgr5Cre^{ER} Apc^{fl/fl}$). Surprisingly we found very few crypts that had recombined Apc, despite being positive for the tdTom reporter (Supplementary Figure 5c). This is not due to leakiness of the reporter since we observed crypts which had partially recombined for Apc, whereas almost the entire crypt is positive for tdTom apart from the Paneth cells which have a longer turnover time (Fig. 4d).

The asynchronous expression of the tdTom reporter and loss of Apc also holds true with a lower induction of 0.15 mg tamoxifen in $Lgr5Cre^{ER} Apc^{fl/fl}$ and with a similar low-level induction in $AhCre^{ER} Apc^{fl/fl}$ mice (Supplementary Figure 5d). Furthermore, serial sections revealed that if the Apc gene is lost, this coincides with accumulation of nuclear β-catenin in the Apc-deficient cells (Supplementary Figure 5e).

To analyse the stem cell dynamics of Apc-deficient cells without use of the tdTom reporter, we analysed $Lgr5Cre^{ER} Apc^{fl/fl}$ mice using a higher concentration of tamoxifen induction (3 mg) by immunohistochemistry for β-catenin. We could differentiate between partially recombined crypts and fully recombined crypts (Fig. 4e), thus allowing us to perform a similar analysis on stem cell dynamics using nuclear β-catenin as a surrogate.

The first detection of nuclear β-catenin+ cell clones was between 4 and 7 days. We observed a shift towards fully recombined crypts from day 4 to day 21 in vehicle-treated mice, suggesting that the β-catenin+ clones replace the WT stem cells and become fixed in the crypt. This shift is accelerated after LGK974 treatment, where at day 7 most of the crypts are fully

populated by β-catenin+ clones (Fig. 4e, f). This confirms our observation in wild-type mice that treatment with the Porcupine inhibitor accelerates the stem cell dynamics. The reduction in partially populated crypts could either result in a fully populated crypt or the removal of these clones from the crypt. Interestingly, when treated with the Porcupine inhibitor, crypts appeared to be fully clonal by day 7—resulting in loss of partial crypts while the number of full crypts remained comparable from day 7 through day 21 (Fig. 4f).

The accelerated fixation time of Apc-deficient crypts was confirmed when we aged $Lgr5Cre^{ER}$ $Apc^{fl/fl}$ mice induced with 3 mg of tamoxifen and treated with Porcupine inhibitor. Here, we observed a dramatic decrease in the time to intestinal tumorigenesis (Fig. 4g). Histological analysis revealed that these mice had numerous intestinal adenomas (Fig. 4h). Interestingly, we observed many lesions in the proximal small intestine, whereas the majority of lesions in control mice were confined to the distal part of the small intestine (Supplementary Figure 6a). This tumour distribution is also reflected in the ratio of Apc deletion. Whereas the tdTom reporter would suggest a higher number of recombined crypts in the proximal compared with the distal small intestine (Supplementary Figure 6b), we observed the opposite in terms of Apc deletion. Analysis of several mice revealed that about 70% of the tdTom+ crypts still expressed Apc based on the RNA in situ probe signal in the proximal small intestine (duodenum). Only a small fraction of crypts (<5%) were positive for the tdTom reporter and had lost Apc exon 14 expression in the majority of cells. In the distal part (ileum) we observed a higher fraction of tdTom+ cells that had also lost Apc (>20%) (Supplementary Figure 6c). This correlates with the high number of adenomas in the distal small intestine compared to the proximal part.

To show that our finding is not simply a reflection of the $Lgr5Cre^{ER}$-mediated deletion we treated $VilCr$ $Apc^{fl/+}$ mice with the Porcupine inhibitor, similarly induced with a low concentration of tamoxifen. Again, we observed a decreased survival of treated mice, whereas control mice showed no signs of intestinal adenoma burden within 100 days p.i (Supplementary Figure 7a). We saw a high frequency of adenomas within the proximal small intestine, whereas no macroscopic adenomas were found in control mice (Supplementary Figure 7b). Microscopic analysis revealed that control mice only had few small lesions, compared to numerous adenomas in mice after treatment with LGK974 (Supplementary Figure 7c).

## Discussion

In summary, we show that the number of ISCs is regulated by secreted Wnt ligands. Reduction of Wnt ligand secretion reduced the stem cell pool and led to a faster fixation of a single ISC clone, probably due to reduced competition for the stem cell niche (Supplementary Figure 8). Despite down-regulation of several stem cell genes (e.g. Lgr5), the cells at the centre of the crypt are functional stem cells and maintain intestinal homoeostasis.

Reducing Wnt ligand secretion led to a reduced number of ISCs. If the ISCs carry a mutation in Apc, this decreased cell competition results in faster population of the crypt by the Apc-deficient cells and hence accelerated tumorigenesis. Importantly we see this across multiple different models of Apc loss (Lgr5 mediated deletion of both copies of Apc, and in $VilCre^{ER}$ $Apc^{fl/+}$ followed by loss of heterozygosity of the remaining wild-type allele).

Our work raises an important technical consideration for previous work looking at selective advantages/disadvantages of

particular alleles using the tdTom reporter. It may be that the problem of overestimating the recombination efficiency is unique to the $Apc^{580S}$ allele in combination with the R26-LSL-tdTom allele. In this case, a total of three alleles need to be recombined to turn on the reporter and achieve homozygous deletion of Apc. However previous studies have shown discordant recombination even if two reporter alleles are at the R26 locus[16].

It should be noted that, as the $Apc^{580S}$ mouse is a hypomorph, any changes in previously published work might be down to the reduced expression of Apc compared to wild-type mice. Moreover, it raises important questions on work suggesting that Apc loss does not lead to Wnt signalling activation and there may be existence of occult Apc-deficient crypts. Using the RNA in situ probes, we do not observe deleted Apc crypts that have not accumulated nuclear β-catenin and the associated phenotypes.

The low efficiency of Apc loss in 3 mg tamoxifen-treated mice is also reflected in the distribution of adenomas in these models. However, we believe the most likely way to explain this phenotype is via differences in the Wnt gradient along the small intestine and maybe partly due to different recombination efficiencies. Previous studies have suggested that the mouse proximal small intestine may have a higher levels of basal Wnt signalling, making it suboptimal for tumorigenesis with certain Apc mutations, e.g. $Apc^{Min/+}$(ref. [23]). Here by using a Wnt inhibitor we make the proximal intestinal more permissive for tumorigenesis following Apc loss, again reinforcing the 'just right' hypothesis of Wnt signalling.

A recent study found a similar role in limiting the number of ISCs by blocking the R-spondin ligands. Inhibition of R-spondins also led to disappearance of Lgr5+ ISCs, despite normal homoeostasis. Similarly the authors observed a shorter time to monoclonality, suggesting that reduction of Wnt ligands or R-spondin can lead to the observed stem cell dynamic changes[13].

One question that arises from our study is why we observed only a minor impact in homoeostasis in long-term porcupine inhibitor-treated intestines. Our current hypothesis is that even reduced Wnt signalling at the base of the crypt is sufficient to maintain epithelial homoeostasis, despite a reduction in the number of ISCs. A recent study in intestinal organoids may also support this paradigm. Here, Wnt3 did not freely diffuse but is bound to the membrane and spreads passively due to cell division[24]. In this case, we could imagine that a reduction of the Wnt ligands bound to the membrane are sufficient for the ISCs in the centre of the niche, but since proliferation is not changed the cells further away receive less Wnt ligands and are therefore lost from the ISC pool. Therefore, further reduction of Wnt signalling with higher doses of Porcupine inhibitor or lowering β-catenin expression can then affect the stem cells in the centre as well, leading to their differentiation and loss of the intestinal crypts.

Our work has also shown that the increase in the number of crypts per circumference after MAPK pathway activation is Wnt ligand dependent. Here, we were able to stop the additional crypt fission by treatment with the Porcupine inhibitor (Supplementary Figure 3). Therefore, Wnt inhibition may prevent tumours developing from more serrated routes, which are often associated with a lack of APC mutation and carry BRAF or KRAS mutations.

## Methods

**Mice and treatment.** All experiments were performed following the UK Home Office guidelines. All mice were maintained under non-barrier conditions and given a standard diet and water ad libitum. The following mouse strains were used:

*VilCre^ER* (ref. [25]), *Lgr5Cre^ER* (ref. [26]), *AhCre^ER* (ref. [27]), *R26R-LSL-tdTomato* (*tdTom^fl*) (ref. [28]), *Apc^fl* (ref. [29]), *Catnb^lox(ex3)* (ref. [30]), *Braf^V600 E* (ref. [31]), *PTEN^fl* (ref. [32]). The *Lgr5Cre^ER Apc^fl/fl* and *VilCre^ER Apc^fl/+* mice were on a C57/B6 background (backcrossed ≥10 generations). The Porcupine inhibitor LGK974 (also referred to as WNT974) was administered in a concentration of 5 mg/kg BID (oral) in a vehicle of 0.5% Tween-80/0.5% methylcellulose. *AhCre^ER* mice were induced with 1 mg β-naphthoflavone (Sigma) and 0.15 mg tamoxifen (Sigma) IP. *VilCre^ER* and *Lgr5Cre^ER* mice were induced with tamoxifen (Sigma) IP at the concentrations indicated throughout the manuscript.

**Immunohistochemistry/RNA in situ hybridisation**. Standard immunohistochemistry techniques were used throughout this study. The following primary antibodies were used: BrdU (1/200, #347580, BD Biosciences), β-catenin (1/50 #610154, BD Biosciences), lysozyme (1/200, DAKO #A0099), RFP (1/200, Rockland #600-401-379). RNA in situ hybridisation (RNAscope) was performed according to the manufacturer's protocol (ACD RNAscope 2.0 High Definition–Brown) for *Lgr5* and *Olfm4*. BaseScope (also ACD) Apc EX14 #701641 (detects wild-type APC exon 14) and Apc E14E16 #703011 (detects floxed APC) were used according to the manufacturer's instructions.

Staining for nuclear β-catenin and RNA in situ hybridisation was performed on tissue samples fixed at 4 °C for less than 24 h in 10% formalin prior to processing.

**RNAseq**. Whole tissue from the small intestine was used for RNA purification. The RNA Integrity was analysed with a NanoChip (Agilent RNA 6000 Nanokit #5067-1511). A total of 2 μg of RNA per sample was purified with Poly-A selection. The count matrix returned from the SAM tools was analysed with the R-package DESeq[33] which returned differentially expressed genes with threshold of the adjusted *p*-value (padj) of <0.1. A heatmap of the significantly deregulated genes was created based on the shape of gene expression by Pearson correlation.

**In vivo imaging**. The in vivo imaging was performed as previously described[1]. *Lgr5Cre^ER R26R-LSL-tdTomato* Mice (*tdTom^fl*) were induced with 0.05 mg tamoxifen. After placing the abdominal imaging window (AIW), mice were kept under anaesthesia and were imaged once a day. After the imaging sessions the mice were allowed to wake up to maintain their body temperature. After imaging, acquired z-stacks were corrected for z and xy shifts using a custom-designed VisualBasic software program and further processed and analysed using basic functions in ImageJ software (linear contrasting, blurring, median filtering).

**Clonal counting**. *Lgr5Cre^ER tdTom^fl* mice were induced with 0.15 mg tamoxifen (IP) as previously described[14]. AhCreER *tdTom^fl* mice were induced with 1 mg β-naphthoflavone and 0.15 mg tamoxifen (IP). The small intestines of mice were sampled at different time points and fixed with freshly prepared 4% paraformaldehyde for 3 h at room temperature. The small intestinal tissue was then incubated with DAPI (10ug/ml) in 0.1% PBS-Tween20 (PBS-T) overnight. Whole mount sections were then imaged using a Zeiss 710 confocal microscope.

**Regeneration**. C57/B6 mice were irradiated with 10 Gy and treated with LGK974/ WNT974 (Porcupine inhibitor) or vehicle 6 h after irradiation. The mice were sampled 72 h after irradiation. The number of regenerating crypts per circumference (10 per mouse) of the small intestine was scored and the average of regenerating crypts per mouse represented in the graph.

**Statistics**. All data were analysed with R[34] and the use of the ggplot2 package[35] and the survival package[36].

**Data availability**. Microarray data that support the findings of this study have been deposited in the ArrayExpress database (www.ebi.ac.uk/arrayexpress) under accession number E-MTAB-4178.

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

## Acknowledgements

The research was supported by Cancer Research UK core grant C596/A17196 and grant to OS A12481, and D.J.H. was funded by the European Union Seventh Framework Programme FP7/2007-2013 under grant agreement number 278568. O.J.S. is funded by an ERC Starting grant COLONCAN under agreement number 311301. This work was financially supported by European Research Council Grant CANCER-RECURRENCE 648804 (to J.v.R.), by the CancerGenomics.nl http://CancerGenomics.nl (Netherlands Organisation for Scientific Research) program (to J.v.R.), by the Doctor Josef Steiner Foundation (to J.v.R) and by the European Union's Horizon 2020 research and innovation program under Marie Sklodowska-Curie grant agreement no. 642866 (to J.v.R).

## Authors contributions:

D.J.H. and O.J.S. contributed to study design; D.J.H., L.B., M.C.H., P.C., A.D.C., D.M.G., M.S.A., C.N., L.B.Z. and R.A.R. to acquisition of data; D.J.H., M.C.H., D.J.W., E.M., L.B., M.E.M., W.J.F., J.v.R., H.J.S. and O.J.S. to analysis and interpretation of the data; and D.J.H., M.C.H. and O.J.S. to drafting the manuscript.

## Additional information

**Competing interests:** M.E.M. is an employee of Novartis Inc. The remaining authors declare no competing interests.

