## [Peer Review File(PDF 448 kb) · Nature Communications]

Editorial Note: parts of this peer review file have been redacted as indicated to maintain the confidentiality of unpublished data.

Reviewers' comments:

Reviewer #1 (Remarks to the Author):

The revised manuscript by Huels et al. has not been changed much from the previous submission. Therefore I still stand with my previous judgement that fig. 1-3 do not present any relevant novelty (as also indicated by rev #3) that warrants publication in a prime research journal. Furthermore, the current interpretation of the key finding that porcupine inhibition increases adenoma formation by reducing stem cell numbers is not convincingly supported by the data. Fig 4d-f clearly show that LGK974 promotes growth of *Apc*^{-/-} cells. However, because the adenomas are mainly larger and not more frequent -as the authors' hypothesis would dictate- suggests another mechanism is in fact responsible for this observation. (Note that only macroscopic lesion counts are presented in Fig 4d, and note the clear difference in lesion size between 4e and 4f). For more specific comments please see my previous comments that have been addressed only remotely.

Reviewer #4 (Remarks to the Author):

I have now read carefully the manuscript by Huels et al. and the comments of both reviewers, with particular attention to those of reviewer #1. My feeling is that reviewer #1 is, generally, correct, as I explain below.

Basically, there are two important findings reported in this study. Firstly, intestinal epithelial stem cells behave differently in response to inhibition of porcupine in their niche, depending on whether they are centrally or peripherally located within the stem cell zone. Secondly, the smaller stem cell population resulting from Wnt pathway inhibition results in crypts that become monoclonal more quickly and confers markedly increased susceptibility to tumorigenesis. In my opinion, if these findings were well demonstrated, they would deserve publication in Nature Communications. However, unfortunately, there are at least two important concerns that must be resolved.

1) The observation that intestinal crypt stem cells behave differently depending on their location is new and potentially important, yet remains descriptive and lacks a convincing explanation. Understanding the molecular basis of the differential susceptibility to porcupine inhibition within the stem cell zone would strongly strengthen the manuscript.

2) As also mentioned by reviewer #1, the causal link between the reduced number of stem cells and the accelerated drift towards monoclonality on the one hand, and the increased tumor initiation on the other, is not well demonstrated.

I would add that a doubt remains as to whether the data have been correctly interpreted. Technically, the experiments of lineage tracing with *Lgr5*-CreER and the RFP reporter, and the experiment of tumorigenesis with *Lgr5*-CreER and *Apc*-flox alleles, are similar: a genetic alteration is made in rare stem cells and the expansion of a cellular clone produced from this altered stem cell is analysed. As such, it is difficult to reconcile the data whereby porc. inhibition results in fewer RFP⁺ clones in one experiment but more tumors in the other. In fact, the reduction in the number of clones observed by RFP genetic tracing (Ext. Data Fig 4A) due to the loss of border-located stem cells in porc. inhibitor-treated mice (Fig 3E, F) suggests that a given genetic modification has a lower probability of being maintained in the crypt in mice treated with the porc. inhibitor. A

possible explanation might be that the systemic porcupine inhibitor treatment impacts upon additional processes that facilitate tumor initiation (e.g. by altering immune surveillance, which might also rely on Wnt activity). Alternatively, the inhibitor might have off-targets that affect tumorigenesis. How can the authors rule this out? Overall, these considerations cast some doubts on the model proposed by Huels et al., and convincing answers are needed before this can be published, possibly involving rescue experiments or genetic inactivation of porcupine. Experiments in organoids, combining the Apc-flox and LSL-RFP reporter, might also help.

Other comments:

I agree with reviewer #1 on the following points:

- A significant first part of the manuscript reports data that recapitulate previously published work, whereas some of the very novel and interesting findings could have been investigated in more detail (see above).

- Fig 2a does not support the statement that the number of clones decreases after porcupine inhibition (point #2). It is important that the authors provide a convincing response to this point. Extended data figure 4a shows a reduction of the number of clones per field but this is not obvious in fig 2a and Ext. Data fig 4b. A solution might be to show low magnification images to demonstrate the decrease of clone number and higher magnification images to illustrate increased clone size in porcupine-inhibited mice.

Having said that, I disagree with reviewer #1 about the dispensable nature of fig 3. These experiments provide direct visualisation of the cellular clones in vivo and therefore very strongly support the conclusions of the different behaviour of stem cells according to their location in the stem cell zone. I find this to be an elegant and technically innovative demonstration. Nevertheless, as mentioned above, I agree that more understanding of the mechanisms underlying these differences would improve the manuscript.

The authors' response to the apparent discrepancy between figures 2c and 3b (point 4 of reviewer #1) seems appropriate to me, as does the response to point 5. I do not completely understand the point 6 of reviewer #1. Points 8 and 9 refer to data that have been removed from the current version and are therefore no longer applicable.

Response to Reviewers' comments:

Reviewers' comments:

Reviewer #1 (Remarks to the Author):

The revised manuscript by Huels et al. has not been changed much from the previous submission. Therefore I still stand with my previous judgement that fig. 1-3 do not present any relevant novelty (as also indicated by rev #3) that warrants publication in a prime research journal. Furthermore, the current interpretation of the key finding that porcupine inhibition increases adenoma formation by reducing stem cell numbers is not convincingly supported by the data. Fig 4d-f clearly show that LGK974 promotes growth of Apc^{-/-} cells.

We would like to argue that porcupine inhibition has no effect on Apc deficient cells. Loss of Apc results in upregulation of the Wnt pathway independent of Wnt ligands. To show this, we specifically performed the experiment of *VilCre Apc^{fl/fl}* mice treated with LGK974. We did not observe any differences in proliferation, as expected. Please see Figure 4A. Therefore we can safely say that porcupine inhibition has no effect on the growth of Apc deficient cells.

However, because the adenomas are mainly larger and not more frequent -as the authors' hypothesis would dictate- suggests another mechanism is in fact responsible for this observation. (Note that only macroscopic lesion counts are presented in Fig 4d, and note the clear difference in lesion size between 4e and 4f).

We analysed with the low level tamoxifen induction adenoma numbers macroscopically and microscopically and observed a clear increase in the number of lesions as well as their size. In addition, even with the 3mg induction we see more adenomas after LGK974 treatment (Figure 4G). Additionally, we quantified b-catenin positive crypts after 3mg tamoxifen induction early after induction. We also observe a clear increase in the ratio of full crypts versus partial crypts and an overall higher number of crypts with b-catenin positive cells (Figure 4 D, E). In summary, all these data strengthen our hypothesis. This reflects an increased propensity for adenomas to now form in the proximal intestine. This is predicted by the "just right" hypothesis. We now discuss this very carefully

For more specific comments please see my previous comments that have been addressed only remotely.

Reviewer #4 (Remarks to the Author):

I have now read carefully the manuscript by Huels et al. and the comments of both reviewers, with

particular attention to those of reviewer #1. My feeling is that reviewer #1 is, generally, correct, as I explain below.

Basically, there are two important findings reported in this study. Firstly, intestinal epithelial stem cells behave differently in response to inhibition of porcupine in their niche, depending on whether they are centrally or peripherally located within the stem cell zone. Secondly, the smaller stem cell population resulting from Wnt pathway inhibition results in crypts that become monoclonal more quickly and confers markedly increased susceptibility to tumorigenesis. In my opinion, if these findings were well demonstrated, they would deserve publication in Nature Communications.

However, unfortunately, there are at least two important concerns that must be resolved.

1) The observation that intestinal crypt stem cells behave differently depending on their location is new and potentially important, yet remains descriptive and lacks a convincing explanation.

Understanding the molecular basis of the differential susceptibility to porcupine inhibition within the stem cell zone would strongly strengthen the manuscript.

During the revision of our paper, two excellent manuscripts were published which help to understand the molecular basis and are complementary to our work. The first manuscript by the Clevers group showed that Wnt ligands can be propagated in a membrane-bound form. Wnt gradients could be generated by cell proliferation and “dilution” of the Wnt-cell-membrane. One could imagine that the intestinal stem cells at the bottom of the crypt receive Wnt ligands and have the highest concentration of Wnt on their membrane. The border stem cells, either don't receive the same amount of Wnt ligands or they derive from center ISCs and the Wnt ligand has already been diluted via cell division. This would explain why particularly the border stem cells are more sensitive to reduced Wnt ligand secretion. The second manuscript by the group of Calvin Kuo postulated a similar decrease in fixation time for single stem cell clones that we observed. Instead of Wnt ligand reduction, they used inhibitors of the R-spondin ligands. Importantly, although R-spondin inhibition resulted in loss of Lgr5 expression, homeostasis and proliferation in the intestinal crypt was unchanged. Therefore, similar to our study, reduction of intestinal stem cells resulted in rapid fixation of the remaining stem cell clones. We have cited both papers and discussed them in the text.

2) As also mentioned by reviewer #1, the causal link between the reduced number of stem cells and the accelerated drift towards monoclonality on the one hand, and the increased tumor initiation on the other, is not well demonstrated.

I would add that a doubt remains as to whether the data have been correctly interpreted.

Technically, the experiments of lineage tracing with Lgr5-CreER and the RFP reporter, and the experiment of tumorigenesis with Lgr5-CreER and Apc-flox alleles, are similar: a genetic alteration is made in rare stem cells and the expansion of a cellular clone produced from this altered stem cell is analysed. As such, it is difficult to reconcile the data whereby porc. inhibition results in fewer RFP+ clones in one experiment but more tumors in the other. In fact, the reduction in the number of clones observed by RFP genetic tracing (Ext. Data Fig 4A) due to the loss of border-located stem cells in porc. inhibitor-treated mice (Fig 3E, F) suggests that a given genetic modification has a lower probability of being maintained in the crypt in mice treated with the porc. inhibitor. A possible

explanation might be that the systemic porcupine inhibitor treatment impacts upon additional processes that facilitate tumor initiation (e.g. by altering immune surveillance, which might also rely on Wnt activity). Alternatively, the inhibitor might have off-targets that affect tumorigenesis. How can the authors rule this out?

We want to highlight that porcupine inhibitor treatment only accelerates the fixation and disappearance of clones in wildtype cells. At day 10 we observe fewer clones and most of these crypts are fully repopulated by RFP cells. The porcupine treated mice reach their final number earlier than vehicle treated mice. We have now included day 30 and day 50. This confirms that there is no significant difference in the number of RFP crypts between vehicle and porcupine inhibitor treated mice at later time points.

That said when we now look in the setting of Apc deficiency we now do see more Apc deficient lesions particularly in the proximal intestine (Ext. Fig. 6B). Again we see a very rapid fixation of Apc deficient cells (Fig. 4D, E) which we believe is the reason for the accelerated tumourigenesis we see. We also see the retention of more b-catenin positive cells (Fig. 4E), which we would expect from our previous studies. This fits well with previous data and the “just right” hypothesis that Wnt gradients favour the formation of tumourigenesis in the distal intestine. This is due to higher Wnt basal levels in the proximal small intestine that in combination with Wnt activation result in excessive Wnt signalling and suboptimal polyp formation ((Leedham et al., 2012)). Due to the porcupine inhibitor treatment we change the basal Wnt levels allowing adenoma formation in the proximal small intestine.

Overall, these considerations cast some doubts on the model proposed by Huels et al., and convincing answers are needed before this can be published, possibly involving rescue experiments or genetic inactivation of porcupine. Experiments in organoids, combining the Apc-flox and LSL-RFP reporter, might also help.

General Response.

We spent considerable time addressing these key concerns and whilst doing this analysis it was clear the reviewer was correct in their interpretation that there seems to be a disconnect between the reporter and the tumour analysis.

We became suspicious of this due the fact that when Apc was deleted at low levels we were seeing very few tumours (although this had been reported in the literature before), despite the tdTom reporter showing high recombination efficacy. Indeed none of the low dose approaches used by the community (e.g (Vermeulen et al., 2013)) led to tumours. This to us seemed at odds to our knowledge that Apc is very efficient at transforming the intestine when higher levels of cre induction is used. We also knew that the conditional APC allele most used by the community is hypomorphic (580s) due to a neomycin cassette remaining in the targeted locus which can give phenotypes in organs such as the liver (Buchert et al., 2010) and therefore one potential hypothesis is that this discord in tumourigenesis at low dose was the fact that the Apc allele recombined much more poorly than the tomato reporter.

Therefore we worked with ACD to develop a “Basescope” probe to allow us to detect the exon deleted (exon 14) in the APC 580S mouse via RNA in situ hybridization. We validated this using highly penetrant deletion and in models that were heterozygous for Apc and developed tumours through LOH. In both of these cases the probes were lost in the recombined cells (Extended Data Figure 5 A). Importantly when we went back to low dose deletion that gives single stem cell with the tomato reporter, there was virtually no recombination of Apc (Extended Data Figure 5 B, C). This should probably not surprise us, as it has been reported previously that targeting of two different reporter loci has very poor concordance even when both were at the ROSA26 locus when low levels of cre induction were used (McCutcheon et al., 2010). So in summary we believed this could explain the differences between the tdTomato clone numbers observed and the low numbers of adenomas seen after induction with the same tamoxifen concentration.

We therefore first determined a minimum concentration of tamoxifen which results in efficient tumourigenesis (Rebuttal Figure 1 and Extended Data Figure 4B).

Figure 1. Tamoxifen Dose for Tumourigenesis Lgr5 Apc fl/fl mice were induced and sampled at 100 days post induction or when sick. All 3mg and one of the 2mg inductions were sampled prior to 100days due to sickness. The number of adenomas/lesions were analysed microscopically.

Therefore we chose a dose of 3mg and developed a scoring system of sections so that we could quantify the number of partially nuclear β -catenin positive crypts relative to fully positive crypts. Given the concordance with probe loss and nuclear b-catenin we used nuclear b-catenin staining to identify the crypt. Interesting, even at 3mg doses most of the crypts were only partially positive for nuclear b-catenin indicating that at this high dose not all ISCs within the crypt have fully lost Apc (Rebuttal Figure 2 and Figure 4 D, E).

Figure 2. Representation of types of β -catenin positive lesions with a 3mg induction of tamoxifen. This highlights that even at a high induction of 3mg recombination of all of the stem cells within a crypt is an unusual event.

Again we saw a dramatic difference of LGK treated mice compared to vehicle treated mice following Apc loss, with very rapid fixation of mutant clones (even by day 10) but also now many more clones retained. Moreover you now also see clones that are not associated with the base of the crypt.

Thus this rapid fixation of clones in the crypt that would have potential to go on to form tumours with the retention of more Apc deficient cells clearly explains the increase in the speed to tumourigenesis we observe and the more lesions in the Lgr5 Apc 3mg mice (Figure 4 F, G). We now also somewhat understand the regional difference we observe in the Lgr5 Apc model. Whereas the tdTom reporter suggests highest recombination efficiency in the proximal small intestine, we generally observe most of the adenomas in the distal part of the small intestine. Analysis with the BaseScope probes confirmed that the proximal SI has the lowest efficacy in terms of Apc loss, which is higher in the distal part. It is interesting to note that most of the stem cell dynamic studies were focussed on the proximal SI, due to the belief that it is the location with the highest recombination. Therefore this helped us understand the second phenotype in the mice with the changed distribution of lesions. Nevertheless, we do believe that this agrees with the 'just-right' amount of Wnt signalling needed for adenoma formation which is also different between the proximal and distal small intestine(Leedham et al., 2012).

We have included the data from the low level deletion of APC still in terms of tumourigenesis as it clearly shows that the very few clones initiated have an increased advantage with the Porcupine inhibitor and now also include the 3mg/kg data as well. We have also added data where we used VillinCreER Apcf1/+ mice. Here loss of the second copy (presumably from the stem cell) will be sporadic. These mice exhibited accelerated tumourigenesis and more proximal lesions were formed in all of these mice.

[Redacted]

Other comments:

I agree with reviewer #1 on the following points:

- A significant first part of the manuscript reports data that recapitulate previously published work, whereas some of the very novel and interesting findings could have been investigated in more detail (see above).
- Fig 2a does not support the statement that the number of clones decreases after porcupine inhibition (point #2). It is important that the authors provide a convincing response to this point. Extended data figure 4a shows a reduction of the number of clones per field but this is not obvious in fig 2a and Ext. Data fig 4b. A solution might be to show low magnification images to demonstrate the decrease of clone number and higher magnification images to illustrate increased clone size in porcupine-inhibited mice.

We included a more representative image for Figure 2A and also moved the clones per field count from the Extended Data to the main figure. Additionally we also include now day 30 and day 50 which show that there is no difference in the final number of crypts with RFP+ cells in the wildtype mice (non Apc mutated mice). It is important to note that the reduced number of clones is only observed at day 10, but not at day 21 and following. This is due to the neutral drift in untreated/vehicle treated mice, which predicts that within the first weeks most of the labelled stem cells will be replaced by non-labelled stem cells. The resulting lower number of clones at day 10 is a mere acceleration in the fixation of single clones.

Having said that, I disagree with reviewer #1 about the dispensable nature of fig 3. These experiments provide direct visualisation of the cellular clones in vivo and therefore very strongly support the conclusions of the different behaviour of stem cells according to their location in the stem cell zone. I find this to be an elegant and technically innovative demonstration. Nevertheless, as mentioned above, I agree that more understanding of the mechanisms underlying these differences would improve the manuscript.

The authors' response to the apparent discrepancy between figures 2c and 3b (point 4 of reviewer #1) seems appropriate to me, as does the response to point 5. I do not completely understand the point 6 of reviewer #1. Points 8 and 9 refer to data that have been removed from the current version and are therefore no longer applicable.

Reviewers' comments:

Reviewer #3 (Remarks to the Author):

A considerable amount of new data has been provided by the authors to answer the referees' comments on the previous version of the manuscript. However, these new pieces of data fail to fully convince and even sometimes add confusion, especially to the last section (adenoma formation). Specific comments on each section below:

1) Wnt inhibition only has minor effects on homeostasis

Minor comments:

- The authors mention previous work from the de Sauvage group "these cells (Lgr5+) are dispensable for homeostasis". The duration for which such a dispensability has been observed should be mentioned.
- In the absence of oncogenic mutations, treatment with LGK974 results in a decreased number of stem cell markers, although some of them are known Wnt target genes (Axin2, Lgr5) and are likely directly affected by Wnt signalling inhibition.
- In the presence of BRaf or BRaf PTEN mutations, porcupine inhibition results in a decreased number of crypts per circumference. I am not sure about the added value of these data in this manuscript.

2) Reduced Wnt ligand secretion decreases crypt fixation time

Major comments:

- The usefulness of the AhCreER experiment (lines 119-124) is not explained clearly.
 - The quality of figure 2 is not sufficient (the legends of the graphs are too small)
- In figure 2E-2F, there is a risk that the clones originate in the low TA compartment and are therefore lost rapidly.
- Although the reduction of clone fixation time during porcupine inhibition is clearly shown, the differential stability of centre versus border stem cells is not fully convincing.

3) Reduced stem cell pool facilitates adenoma formation

Major comments:

- The authors provide evidence that porcupine inhibition facilitates tumorigenesis. However, the mechanisms involved remain poorly understood. Furthermore, this section has become very complex and should be rewritten more concisely.
- Even the initial statement that because porcupine inhibition results in an acceleration of crypt clonality, LGK974 should accelerate tumorigenesis is arguable. Indeed, a decreased number of stem cells in LGK974-treated mice implies a reduction of the target stem cell pool for tumorigenesis. How these two parameters (less target cells and accelerated clonality) will influence the frequency of adenoma formation is unclear, and computer modelling may help integrating the relative impact of each parameter on the observed phenotype.
- Using the Basescope technology is certainly a good idea but the characterization of the method shown in Ext Fig 5 is not convincing, with nearly invisible stainings, insufficient insets and arrows to understand the pictures.

Overall, many important questions remain unsolved. In particular, critical controls are missing to state that homeostasis is not or very little affected by porcupine inhibition. For instance, the reduction in stem cell numbers may cause cellular plasticity, with TA cells acquiring stem cell properties. How would such a regeneration program affect tumorigenesis? Are stem cells, as identified in this study, the only cells to be able to initiate adenomas (not talking about the alternative models combining multiple genetic mutations)? If TA cells play a role in adenoma formation, is there a differential dependence upon Wnt signalling between CBC stem cells and TA cells? Would such TA-of-origin cells also be subject to clonal selection? If not, would this explain at least partially the increased adenoma initiation rate during porcupine inhibition?

Response to Reviewers' comments:

Reviewers' comments:

Reviewer #3 (Remarks to the Author):

A considerable amount of new data has been provided by the authors to answer the referees' comments on the previous version of the manuscript. However, these new pieces of data fail to fully convince and even sometimes add confusion, especially to the last section (adenoma formation). Specific comments on each section below:

1) Wnt inhibition only has minor effects on homeostasis

Minor comments:

- The authors mention previous work from the de Sauvage group "these cells (Lgr5+) are dispensable for homeostasis". The duration for which such a dispensability has been observed should be mentioned.

We thank the reviewer for the comment. The Lgr5 cell dispensability was measured over 10 days in the study of the de Sauvage group. The reason why mice were not aged longer was due to liver toxicity caused by loss of Lgr5 cells in this tissue. We have now mentioned this in the text.

- In the absence of oncogenic mutations, treatment with LGK974 results in a decreased number of stem cell markers, although some of them are known Wnt target genes (Axin2, Lgr5) and are likely directly affected by Wnt signalling inhibition.

We agree with this point but also highlight that non-Wnt stem cell target genes (Olfm4) are also reduced.

- In the presence of BRAF or BRAF PTEN mutations, porcupine inhibition results in a decreased number of crypts per circumference. I am not sure about the added value of these data in this manuscript.

We feel that these data provide an interesting and valuable contribution to the field of porcupine inhibitors, particularly over the difference between hyperplasia and homeostasis. However we are happy to remove these data if the editor would like us to.

2) Reduced Wnt ligand secretion decreases crypt fixation time

Major comments:

- The usefulness of the AhCreER experiment (lines 119-124) is not explained clearly.

We have edited this section to make it easier to interpret the importance of this experiment. Most importantly it shows that the observed change in stem cell dynamics is independent of the Cre system used.

- The quality of figure 2 is not sufficient (the legends of the graphs are too small)

We thank the reviewer for this observation and agree that the legends were too small. We have now altered these legends to make them easier to read.

In figure 2E-2F, there is a risk that the clones originate in the low TA compartment and are therefore lost rapidly.

We believe the reviewer is most likely referring to Figure 3E-F. Using the Lgr5CreER system we would not be able to label cells in the low TA compartment since the cre itself is limited to recombination in the Lgr5

positive stem cells. If however the labelled cells originated from the boarder stem cell zone (which abuts the TA compartment) these cells would then be more likely to be lost. This is shown in the figure (centre versus border stem cells).

- Although the reduction of clone fixation time during porcupine inhibition is clearly shown, the differential stability of centre versus border stem cells is not fully convincing.

We respectfully disagree with the reviewer regarding these data. We strongly believe that the difference between the centre and boarder cells is clearly displayed when you compare figure 3D to figure 3F (ie no difference in the loss of centre stem cells with Porcupine inhibitor 3D whilst a large and significant difference in the loss of the border stem cells 3F

3) Reduced stem cell pool facilitates adenoma formation

Major comments:

- The authors provide evidence that porcupine inhibition facilitates tumorigenesis. However, the mechanisms involved remain poorly understood. Furthermore, this section has become very complex and should be rewritten more concisely.

We made changes to the text and additionally provided a schematic to help understand this complex topic (Extended Data Figure 8). We do want to highlight that our finding that the tdTomato reporter was not usable to report loss of Apc did need to be explained.

- Even the initial statement that because porcupine inhibition results in an acceleration of crypt clonality, LGK974 should accelerate tumorigenesis is arguable. Indeed, a decreased number of stem cells in LGK974-treated mice implies a reduction of the target stem cell pool for tumorigenesis. How these two parameters (less target cells and accelerated clonality) will influence the frequency of adenoma formation is unclear, and computer modelling may help integrating the relative impact of each parameter on the observed phenotype.

We have now removed this initiation statement

To clarify for the reviewer, for all of the porcupine treatment experiments we induced recombination first and THEN treated with the porcupine inhibitor. We now make note of this in text and the schematic will help to understand

We also point out that the number of cells with loss of Apc are similarly at early timepoints. "This analysis also confirmed that an equal number of crypts were recombined before the start of the treatment (**Figure 2 E, day 4**)." Therefore for the Lgr5CreER Apc^{fl/fl} experiments with low and high level induction we did not reduce the number of target cells, but induce the oncogenic hit and then reduce the number of stem cells.

- Using the Basescope technology is certainly a good idea but the characterization of the method shown in Ext Fig 5 is not convincing, with nearly invisible stainings, insufficient insets and arrows to understand the pictures.

We apologise for the low quality of the images and provided much better images to highlight the dissymmetry of the tdTom reporter and loss of Apc, (Figure 4D and new Extended Data Figure 5).

Overall, many important questions remain unsolved. In particular, critical controls are missing to state that homeostasis is not or very little affected by porcupine inhibition.

We disagree with the reviewer on this point, we have performed an analysis of proliferation, stem cell and differentiation in the normal intestine both immediately and at 30 days. These are accepted assays in the field to look at homeostasis. Moreover our work is complementary to previous studies which show low concentration of the porcupine inhibitor LGK974 has little effect on intestinal homeostasis¹.

For instance, the reduction in stem cell numbers may cause cellular plasticity, with TA cells acquiring stem cell properties. How would such a regeneration program affect tumorigenesis?

As mentioned in the point above, if the TA cells would acquire stem cell properties, we would expect a reduction in clone size, since only Lgr5 cells recombined with the tomato reporter would be traced and then followed. If TA cells took over that were not Lgr5 positive, crypts would be repopulated from non-labelled cells so it would look like clones are lost (smaller clone size and fewer crypts fully recombined) so the exact opposite of what we see.

Furthermore, although the Lgr5 stem cells lose expression of Lgr5 after LGK treatment, we have shown those at the bottom of the crypt are still functional and contribute to the increase in clone size (Figure 3D). Moreover, the ability of TA cells to acquire stem cell properties has been only observed during intestinal regeneration (whole body irradiation) or when Lgr5 cells were depleted by diphtheria toxin^{2,3}. Such regeneration is accompanied by major morphological changes of the crypt or increase in apoptosis, which has not been observed during this study.

Are stem cells, as identified in this study, the only cells to be able to initiate adenomas (not talking about the alternative models combining multiple genetic mutations)? If TA cells play a role in adenoma formation, is there a differential dependence upon Wnt signalling between CBC stem cells and TA cells? Would such TA-of-origin cells also be subject to clonal selection? If not, would this explain at least partially the increased adenoma initiation rate during porcupine inhibition?

Here we fully agree with the reviewer that these are important points for a study focussing on the possibility of non-stem cells as the cell of origin for adenomas. Indeed we have previously published on the ability of non-stem cell to form tumours if they acquire additional mutations to Apc loss. However we believe a more thorough characterisation of this phenotype is outside the scope of this.

Reference

1. Kabiri, Z. *et al.* Stroma provides an intestinal stem cell niche in the absence of epithelial Wnts. *Development* **141**, 2206–15 (2014).
2. Ritsma, L. *et al.* Intestinal crypt homeostasis revealed at single-stem-cell level by in vivo live imaging. *Nature* **507**, 362–365 (2014).
3. van Es, J. H. *et al.* Dll1(+) secretory progenitor cells revert to stem cells upon crypt damage. *Nat. Cell Biol.* **14**, 1099–104 (2012).